# The MicroRNA Ame-Bantam-3p Controls Larval Pupal Development by Targeting the Multiple Epidermal Growth Factor-like Domains 8 Gene (*megf8*) in the Honeybee, *Apis mellifera*

**DOI:** 10.3390/ijms24065726

**Published:** 2023-03-17

**Authors:** Jing Yu, Hongyu Song, Hongfang Wang, Ying Wang, Zhenguo Liu, Baohua Xu

**Affiliations:** College of Animal Science and Technology, Shandong Agricultural University, Tai’an 271000, China

**Keywords:** honeybee, 20E signaling pathway, microRNA, 20-hydroxyecdysone, ame-bantam-3p, insect development

## Abstract

20-Hydroxyecdysone (20E) plays an essential role in coordinating developmental transitions in insects through responsive protein-coding genes and microRNAs (miRNAs). However, the interplay between 20E and miRNAs during insect metamorphosis is unknown. In this study, using small RNA sequencing, a comparative miRNA transcriptomic analysis in different development stages, and 20E treatment, we identified ame-bantam-3p as a key candidate miRNA involved in honeybee metamorphosis. Target prediction and in vitro dual-luciferase assays confirmed that ame-bantam-3p interacts with the coding region of the *megf8* gene and promotes its expression. Meanwhile, temporal expression analysis revealed that the expression of ame-bantam-3p is higher in the larval stage than in prepupal and pupal stages, and that this expression pattern is similar to that of *megf8*. In vivo, we found that the mRNA level of *megf8* was significantly increased after the injection of ame-bantam-3p agomir. A 20E feeding assay showed that 20E downregulated the expression of both ame-bantam-3p and its target gene *megf8* on larval days five, six, and seven. Meanwhile, the injection of ame-bantam-3p agomir also reduced the 20E titer, as well as the transcript levels of essential ecdysteroid synthesis genes, including *Dib, Phm, Sad, and Nvd*. The transcript levels of 20E cascade genes, including *EcRA*, *ECRB1*, *USP*, *E75*, *E93*, and *Br-c*, were also significantly decreased after ame-bantam-3p agomir injection. However, ame-bantam-3p antagomir injection and *dsmegf8* injection showed the opposite effect to ame-bantam-3p agomir injection. Ame-bantam-3p agomir treatment ultimately led to mortality and the failure of larval pupation by inhibiting ecdysteroid synthesis and the 20E signaling pathway. However, the expression of 20E signaling-related genes was significantly increased after *megf8* knockdown, and larvae injected with *dsmegf8* showed early pupation. Combined, our results indicate that ame-bantam-3p is involved in the 20E signaling pathway through positively regulating its target gene *megf8* and is indispensable for larval–pupal development in the honeybee. These findings may enhance our understanding of the relationship between 20E signaling and small RNAs during honeybee development.

## 1. Introduction

MicroRNAs (miRNAs or miRs) constitute a class of endogenous, non-coding, single-stranded RNAs ranging from 21 to 24 nucleotides in length. They can regulate gene expression post-transcriptionally through binding to target sites in the 5′untranslated region (UTR), coding region, or 3′UTR of target mRNAs [1,2]. The role of miRNAs in the regulation of gene expression was initially thought to be repressive; however, it has been shown that miRNAs also play a positive regulatory role in gene expression [3,4]. For example, miR-466l binds to AU-rich elements in IL-10 mRNA, which in turn increases the expression of IL-10 [5]. During amino acid starvation, miR-10a binding to the 5′ end of ribosomal proteins increased their translation efficiency, whereas inhibition of miR-10a caused the opposite effect [6]. Studies have shown that miRNAs play an important role in regulating insect development, including oogenesis, embryogenesis, molting and metamorphosis [7,8,9]. For example, using RNA interference (RNAi) to silence Dicer-1, a key component in miRNA processing, Gomez-Orte and Belles (2009) found that miRNAs are involved in metamorphosis regulation of the German cockroach in *Blattella germanica*. MiRNAs are conserved among animals [10,11]. Bantam was the first miRNA discovered in *Drosophila melanogaster* [12], and bantam was found to promote systemic growth largely by repressing the production of 20E [13,14]. In addition, bantam has been reported to play an important role in regulating cell survival, proliferation, migration, organ growth during growth, and development during development [15,16,17,18,19,20,21]. Inhibition of bantam activity can lead to effects, such as reduced imaginal tissue growth or impaired dendrite development, which may affect the timing of metamorphosis [22,23]. Studies on imaginal tissues have indicated that, during development, bantam acts as an effector of several signaling pathways, including the Hippo [19,24], Notch [12], decapentaplegic (Dpp) [25], and epidermal growth factor receptor (EGFR) pathways [26].

Multiple EGF-like domain 8 (*MEGF8*) encodes a multi-domain protein conserved in many metazoan species that functions to traffic membrane-bound receptor molecules to the cell surface or to lysosomes for degradation [27,28]. In *Drosophila*, *dMegf8* is involved in mesoderm development [29], and the phenotype of *dMegf8* larval mutants is similar to that of some bone morphogenetic protein (BMP) signaling mutants, such as impaired formation of denticles and bristle [30]. Although genetic screening reports suggest a potentially important role for *dMegf8* in various developmental and behavioral pathways, relatively little is known about it.

20E is a steroid hormone essential for insect development that triggers every major developmental transition in the insect life cycle, including egg hatching, larval–larval molting, and larval–pupa–adult metamorphosis [31,32,33,34]. 20E, the active ecdysteroid, binds to a heterogeneous receptor consisting of the ecdysteroid receptor (EcR) and Ultraspiracle (USP) to activate the expression of many early response genes, including genes encoding transcription factors *E74*, *E75* and *Broad-Complex* [35,36,37]. The stability of gene expression in the 20E signaling pathway is essential for maintaining normal larval–pupal development in insects. Emerging evidence suggests that some non-coding RNAs (ncRNAs) are also involved in regulating the 20E signaling pathway. Specifically, it was shown that miRNAs are involved in the regulation of ecdysteroid and JH, and they even act as nodes in the 20E signaling pathway. In *Drosophila*, 20E regulates the expression of miRNAs involved in the 20E signaling pathway (let-7, miR-100, miR-34, and miR-125) [38,39,40]. Several miRNAs are components of additional systemic signaling pathways in insects, including 20E, insulin, stress, immune, and circadian pathways [41]. Bantam and four other miRNAs (miR-80, miR-89, miR-154, and miR-257) are involved in regulating insect metamorphosis by targeting *spook* and *spookier* in the ecdysteroid synthesis pathway, among which bantam promotes systemic growth by inhibiting the synthesis of the steroid hormone 20-hydroxyecdysone [13,42]. Furthermore, although there is increasing evidence supporting a direct or indirect relationship between miRNA regulation and hormone-related pathways in other insects, it remains unclear whether such a relationship also exists between miRNAs and 20E signaling pathways in honeybees. Here, to determine the function of miRNAs involved in metamorphosis and identify which miRNAs are regulated by 20E in vivo, we fed exogenous 20E to 2-day-oldworker bee larvae to disrupt the normal 20E pulse, and then we performed small RNA sequencing. A conserved miRNA, ame-bantam-3p, was identified as a negatively regulated target of 20E. Employing a dual-luciferase reporter assay and ame-bantam-3p agomir and antagomir injection experiments, we found that *megf8* is a target gene of ame-bantam-3p. The injection experiment results also showed that ame-bantam-3p plays an important role in positively regulating the levels of the *megf8* gene, thus contributing to the homeostasis of 20E signaling pathway-related genes during larval–pupal development. This study constitutes a basis for further investigation of the molecular mechanisms involved in the biogenesis and functions of 20E-regulated miRNAs and may also serve to expand our understanding of how miRNAs participate in insect metamorphosis.

## 2. Results

### 2.1. Analysis of Small RNA Deep-Sequencing Data

To investigate the relationship between 20E signaling pathways and small RNAs during honeybee development and metamorphosis, small RNA libraries were constructed for sequencing from pooled samples of five-day-old larvae, six-day-old larvae, seven-day-old larvae, eight-day-old larvae, and Pw after 20E treatment (see Methods; Appendix A). Between 13.9 and 35.9 million single-end raw reads were obtained from the 29 small RNA libraries generated from the developmental stages analyzed after 20E treatment. After removing low-quality reads, adaptor sequences, and sequences with lengths of <18 nt and >30 nt, approximately 9.7 to 16.0 million clean reads were obtained and were used to predict known and novel miRNAs. The Q30 percentage per individual ranged from 95.76% to 98.49% (Appendix A). rRNAs, tRNAs, snRNAs, snoRNAs, other ncRNAs, and repeat sequences were filtered out, and 43.98% to 76.41% of genome-matched reads were screened out from the total reads (Appendix A). The lengths of mature miRNAs ranged from 18 to 30 nt and one peak at approximately 22 nt was observed for all genome-matched reads, which is considered the classical length of miRNAs. Furthermore, 23.55% to 70.83% of the clean 22-nt reads were predicted to be miRNAs (Appendix A). For the 10 libraries, both known and novel miRNAs with longer sequences exhibited clear uridine (U) bias at the first nucleotide. Meanwhile, analysis of the frequency of bases at each position in all the miRNA reads showed that U and guanine (G) occurred more frequently than adenine (A) and cytosine (C) (Appendix A).

### 2.2. Identification of Differentially Expressed miRNAs

First, using miRBase (v22) and the reference honeybee genome (Apis_mellifera.Amel_4.5_ncbi), we identified 297 miRNAs in the library from the five-day-old larvae, 227 in the library from the six-day-old larvae, 264 in the library from the seven-day-old larvae, 253 in the library from the eight-day-old larvae, and 265 in the library from Pw (Appendix A). After the removal of redundant reads, a total of 424 miRNAs were obtained from all samples, 244 of which were predicted to be known miRNAs and 180 novel miRNAs (Appendix A). To understand the functional characteristics of these predicted miRNAs, we performed an analysis of miRNA families to identify which miRNAs were highly conserved. Some of the miRNAs were found to belong to families, such as the miR-9, miR-252, miR-279, and bantam families, among others (Appendix A), suggesting that they may have conserved roles in the regulation of metamorphosis in honeybees.

The expression levels of miRNAs in each sample were calculated and normalized based on the TPM algorithm [43]. To reveal the functions of the identified miRNAs during honeybee metamorphosis, we compared miRNA expression levels between the 20E and control treatment groups using DESeq2 with |log2(FC)| ≥ 0.58 and *p*-value ≤ 0.05 as the significance cutoffs [44]. Pearson’s correlation analysis showed that there was a high degree of correlation among the samples in each group, indicating that the sampling was reasonable (Appendix A). Although their expression was detected in all libraries, the expression levels of several miRNAs were significantly altered at different developmental stages after 20E treatment. In five-day-old larvae, 39 miRNAs showed significant responses, including 19 that were upregulated and 20 that were downregulated. In six-day-old larvae, 26 miRNAs showed a significant response to 20E, 19 of which were significantly upregulated and 20 significantly downregulated. In seven-day-old larvae, 28 miRNAs displayed a significant response to 20E, with 13 exhibiting significant upregulation and 15 significant downregulation. However, 11 miRNAs in eight-day-old larvae showed a significant response (three upregulated, eight downregulated), four miRNAs showed a significant response in white eye pupae (two upregulated, two downregulated showed significant responses (Table 1, Figure 1); accordingly, we next focused on five-day-old, six-day-old, and seven-day-old larvae after 20E treatment. 

Hierarchical clustering analysis based on the TPM method showed that larvae of the same day old treated with 20E were clustered in the same clade and separated from the respective controls (Figure 2A–E). Furthermore, some miRNAs, such as ame-bantam-3, miR-276b-3p, ame-let-7-5p, and ame-miR-282-5p, were differentially expressed at different developmental stages (Appendix A), indicating that these miRNAs may be involved in the larval to pupal transition of honeybees. Venn diagram analysis showed that the levels of ame-bantam-3p, miR-279b-3p, and miR-6001-5p were significantly decreased in five-day-old, six-day-old, and seven-day-old larvae after 20E treatment, indicating that these miRNAs were negatively regulated by 20E.Ame-bantam-3p was found to be the most abundant miRNA in all three developmental stages. Meanwhile, the expression of some miRNAs (e.g., ame-miR-7-5p, ame-miR-13b-3p, ame-miR-993-3p, and ame-miR-263a-5p) was upregulated in five-day-old and six-day-old larvae after 20E treatment (Figure 2F). The expression patterns of these miRNAs suggested that they may play a significant role during larval–pupal transformation and metamorphosis in the honeybee and that 20E regulates the expression of metamorphosis-associated miRNAs. Based on all the above results, we subsequently focused on ame-bantam-3p, and we examined its function during honeybee metamorphosis.

To ensure the reliability of the small RNA deep-sequencing data and compare the expression profiles of some of the miRNAs among the different developmental stages after 20E treatment, we selected eight differential expression (DE) miRNAs that were highly expressed in one of the five developmental stages and confirmed their expression levels via RT-qPCR. As shown in the histogram in Figure 3, the expression trends of all the tested miRNAs were consistent with the small RNA deep sequencing data, suggesting that the latter were highly reliable.

### 2.3. Prediction and Functional Enrichment Analysis of the Target Genes of DE miRNAs

To characterize the regulatory roles of the identified miRNAs during larval–pupal transformation and metamorphosis in the honeybee, the potential targets of the miRNAs were predicted using miRanda [45] and targetScan [46] software. A total of 161 target genes were predicted for 43 DE miRNAs between control and 20E in five-day-old larvae, and these were successfully annotated against the COG, NR, Swiss-Prot, GO, KEGG, Pfam, and eggNOG databases (Appendix A). KEGG pathway enrichment analysis showed that the target genes of the DE miRNAs were mostly associated with ECM-receptor interaction, Hippo signaling pathway, MAPK signaling pathway, Dorso–ventral axis formation, lysine degradation, and FoxO signaling pathway, suggesting that these genes may play important roles in honeybee metamorphosis (Figure 4). The most abundant terms relating to biological process, cellular component, and molecular function in the GO term enrichment analysis of the predicted target genes are shown in Figure 5. The most significantly enriched terms in the biological process category included homophilic cell adhesion via plasma membrane adhesion molecules, instar larval or pupal development, muscle attachment, PCD involved in cell development, and skeletal muscle organ development. For the cellular component category, the most significantly enriched terms included axoneme, collagen trimer, dynein complex, mitochondrial matrix, and plasma membrane-bounded cell projection cytoplasm. Meanwhile, the most significantly enriched terms in the molecular function class included act in binding, calcium ion binding, extracellular matrix structural constituent, molecular function, and transcription regulator activity.

### 2.4. In Vitro Validation of Ame-Bantam-3p Targets

To identify the target genes of ame-bantam-3p, two miRNA target-prediction software programs, miRanda and TargetScan, were simultaneously employed, leading to the identification of only one putative target gene, *megf8*. Based on the predicted results, miRNA/mRNA combinations with low energy and high score were selected. The putative target sites in the *megf8* coding sequence and the free energy value between ame-bantam-3p and *megf8* mRNA is shown in Figure 6A. In addition, ame-bantam-3p is highly conserved in insects and the honeybee (Figure 6B). To further assess the relationship between ame-bantam-3p and its target gene, we conducted an in vitro dual-luciferase reporter assay. The approximately 400 bp fragment, which covered the bantam-3p binding site in *megf8*, was cloned into a luciferase reporter plasmid and co-transfected into HEK-293 cells with the ame-bantam-3p mimic. A null target sequence upstream of the luciferase coding sequence served as a NC. The results showed that luciferase reporter activity was increased by approximately 41% when the pGL3-CMV-LUC-*megf8* (LOC725895)-CDS was co-transfected with the ame-bantam-3p mimic compared with when *megf8* was co-transfected with the NC (Figure 6C). Together, these results confirmed that there is a functional target of ame-bantam-3p in the coding region of *megf8*, and *megf8* is a target gene of am-bantam-3p.

To further determine whether ame-bantam-3p regulates *Megf8* expression in vivo, we injected either ame-bantam-3p agomir (overexpression) or ame-bantam-3p antagomir (knockdown) into six-day-old larvae. Since neither NC-agomir nor NC-antagomir had a significant effect on the larvae after injection in all measured indices, we used NC-agomir as a control for further analysis in the following experiments. As shown in Figure 6D, the ame-bantam-3p expression level was significantly upregulated in worker bee larvae injected with agomir, whereas antagomir treatment resulted in a significant downregulation of ame-bantam-3p expression. The results showed that, compared with the control group, the mRNA level of *megf8* was increased by approximately 86% following ame-bantam-3p agomir injection, whereas antagomir injection resulted in a 50% decrease in *megf8* expression levels (Figure 6E), which was consistent with the results of the dual-luciferase assay. These findings further indicated that ame-bantam-3p positively regulates *megf8* expression in vivo by binding to the *megf8* coding region.

### 2.5. Ame-Bantam-3p and megf8 Are Co-Expressed in the Larval–Pupal Stages

To investigate the potential function of ame-bantam-3pduringhoneybee metamorphosis, the temporal expression levels of ame-bantam-3p and *megf8* in the larval–pupal stages were assessed by RT-qPCR. Our results showed that ame-bantam-3p expression rapidly increased during the larval stages, peaking at larval day 5, and then it gradually declined. At the prepupal stage (Pp2 to Pp3), ame-bantam-3p expression was almost undetectable (Figure 7A). In the larval stage, the mRNA level of *megf8* gradually increased, followed by a decrease after the prepupal stage, which was consistent with the ame-bantam-3p expression pattern. Focusing on five-day-old larvae, we measured ame-bantam-3p and *megf8* transcript levels in the midgut, fat body, head, epidermis, and silk gland. The results showed that ame-bantam-3p was ubiquitously expressed in all investigated tissues, with the highest relative expression levels being observed in the head, epidermis, and fat body. Similar results were seen for *megf8* expression (Figure 7B,C). The consistency in expression mode between ame-bantam-3p and *megf8*, both in the larval–pupal stages and isolated tissues, showed that ame-bantam-3p might be involved in regulating the development of worker bee larvae by positively regulating its target gene *megf8*.

Next, the levels of 20E were measured in the larval–pupal developmental stages. We found that the 20E titer was negatively correlated with the expression of ame-bantam-3 and *megf8*, remaining at a very low level in the larval stages and at a high level in the prepupal stage (Figure 7D). These observations further confirmed that ame-bantam-3p might have a negative impact on 20E production in the honeybee by targeting the *megf8* gene.

### 2.6. Ame-Bantam-3p Inhibits Ecdysteroid Production through Upregulating Target Gene megf8

To elucidate the effect of 20E oname-bantam-3p and its target gene *megf8*, we measured the expression levels of ame-bantam-3p and *megf8* in five-day-old, six-day-old, and seven-day-old larvae fed exogenous 20E. Compared with the control group, the expression of ame-bantam-3p was downregulated by approximately 60%, 50%, and 34% in, respectively, five-day-old, six-day-old, and seven-day-old larvae fed 20E (Figure 8A). Meanwhile, there was a significant decrease in *megf8* expression levels between the 20E treatment group and the control group (Figure 8B). This 20E feeding assay demonstrated that 20E can suppress the transcription of ame-bantam-3p and that of its target gene *megf8*.

That ame-bantam-3p and 20E have opposing effects on systemic growth suggests that the function of ame-bantam-3p during honeybee development is, at least in part, to inhibit ecdysteroid production. To test this possibility, we injected ame-bantam-3pagomir and ame-bantam-3p antagomir into the hemolymph of six-day-old larvae, respectively, and measured the circulating levels of 20E. We found that agomir-injected larvae failed to molt to white pupae (Figure 8C), and their mortality rate was approximately 12% higher than that of NC-agomir-injected larvae (Figure 8D). However, antagomir injection had no significant effect on larval mortality, but significantly reduced the pupation time of larvae (Figure 8E). This may have been due to interference with ecdysteroid biosynthesis and 20E signaling, ultimately affecting larval pupation. Compared with following NC-agomir injection, the level of 20E was significantly increased in larvae 24 h after agomir injection, while 20E levels were significantly lower in antagomir-injected larvae (Figure 8F). We further evaluated the expression levels of the *Dib*, *Phm*, *Sad*, *Nvd,* and *Shd* genes, which encode enzymes required for ecdysteroid biosynthesis, by RT-qPCR and found that the expression levels of *Dib*, *Phm*, *Sad*, and *Nvd* were lower in agomir-treated larvae than in larvae injected with NC-agomir, in contrast to that seen with antagomir treatment (Figure 8G). This suggested that ame-bantam-3p could negatively affect ecdysteroid biosynthesis. 20E signaling was also lower in the agomir treatment group than in the NC-agomir treatment group, as determined by the expression levels of genes encoding the nuclear receptors *ECRA*, *ECRB1*, and *USP* and those of 20E primary response genes (i.e., the transcription factor-encoding genes *Br-C*, *E75*, and *E93*). In contrast, the expression levels of 20E signaling genes, including *ECRA*, *ECRB1*, *USP*, *E93*, *E75,* and *Br-C* were significantly increased in antagomir-injected larvae compared to NC-injected larvae (Figure 8H). 

We further assayed the proliferation of fat body cells in larvae injected with ame-bantam-3p agomir or am-bantam-3p antagomir. A greater number of proliferating fat body cells was observed in ame-bantam-3p agomir-injected larvae than in larvae treated with NC agomir, as detected by EdU staining, whereas there was no significant effect in the larvae injected with ame-bantam-3p antagomir (Figure 8I). The levels of programmed cell death (PCD) were also detected using TUNEL staining in the fat body. Compared with NC agomir, the apoptosis was not observed at 24 h after ame-bantam-3p agomir injection (Figure 8I), whereas apoptotic signals were greater in larval fat body cells after ame-bantam-3p antagomir injection. In summary, our findings suggested that high ame-bantam-3p levels in young larvae contribute to the maintenance of low 20E titers and the promotion of systemic growth, whereas reduced ame-bantam-3p activity in the prepupal stage contributes to the generation of the 20E peak and the cessation of growth.

To investigate the role of the target gene *megf8* in the regulation of ecdysteroid synthesis by bantam-3p, we injected *dsmegf8* into the hemocoel of six-day-old honeybee larvae and examined the expression levels of genes related to ecdysteroid synthesis. Compared with *dsGFP* injected larvae, mortality of larvae after *dsmegf8* injection was not significantly affected, but it caused early pupation of larvae (Figure 9A–C). The 20E titer and the expression levels of *Dib*, *Phm*, *Sad*, *Nvd* and *Shd* were significantly increased in *dsmegf8*-injected larvae compared to *dsGFP*-injected larvae (Figure 9D and Figure 8E). The expression of several key genes in the 20E-triggered transcriptional cascade was measured in *dsmegf8*-injected larvae. Of the six genes tested, *ECRA*, *ECRB1*, *USP,* and *E75* showed an 80–90% increase in mRNA levels compared to control larvae (Figure 9F). These data suggest that bantam-3p inhibits ecdysteroid synthesis and interferes with 20E signaling by upregulating the expression of the target gene *megf8* during metamorphosis in honeybee larvae. These observations constitute additional evidence that ame-bantam-3p and *megf8* may be involved in larval-pupal development in the honeybee through the regulation of 20E levels.

## 3. Discussion

The steroid hormone 20E triggers significant developmental changes in insects, including molting and metamorphosis. The functions of hormones might be achieved through the interaction of genes involved in hormone signaling with miRNAs that control complex gene networks directing developmental decisions [47,48]. In this study, we investigated the complex relationship between 20E and miRNAs during honeybee metamorphosis by systematically examining differential miRNA expression after 20E treatment and at different developmental stages (from the larval to the pupal stage) and screening for conserved miRNAs. We identified the conserved ame-bantam-3p as a candidate miRNA potentially involved in honeybee metamorphosis, and we explored its role in the regulation of ecdysteroid synthesis and the 20E signaling pathway during larval–pupal development in the honeybee. In recent years, numerous studies have demonstrated that miRNAs can activate the expression of the target genes, such as that shown for miR-373, which can bind to the promoters of E-calmodulin and C2, a cold excitation domain-containing protein, thereby upregulating their expression [49]. In specific conditions, micoRNA-10a binds to ribosomal protein 5’-UTR to activate the translation of ribosomal proteins and ultimately promote total protein synthesis [6]. In the present study, the complete *megf8* gene sequence was subjected to binding site analysis using target-prediction software, leading to the identification of ame-bantam-3p-binding sequences in the coding region of *megf8*. Using a luciferase assay, we further found that ame-bantam-3p positively regulates *megf8* expression through its target site in the coding region of *megf8* (Figure 6B), while bioinformatic analysis and a dual-luciferase reporter assay confirmed the functionality of the target site. Studies have shown that antagomir treatment can markedly reduce miRNA expression levels in insects [50,51,52,53,54]. In this study, ame-bantam-3p agomir treatment upregulated ame-bantam-3p expression, whereas ame-bantam-3p antagomir exerted a significant inhibitory effect on the expression of bantam-3p. The expression level of *megf8* was significantly upregulated in larvae 24 h after ame-bantam-3p agomir injection compared to NC-agomir-injected larvae, while ame-bantam-3pantagomir injection elicited the opposite effect. These results suggest that bantam-3p promotes the expression of *megf8* by binding to its coding region.

In our study, temporal expression analysis indicated that ame-bantam-3p expression rapidly increased in the larval stage, peaked at five-day-old larval, and then gradually decreased, which was consistent with the *megf8* expression pattern. Additionally, we measuredame-bantam-3p and *megf8* transcript levels in the midgut, fat body, head, epidermis, and silk gland, and found that ame-bantam-3p was ubiquitously expressed in all investigated tissues, with the highest relative expression levels being observed in the head, epidermis, and fat body. Importantly, similar results were seen for *megf8* expression (Figure 7). Combined, these findings suggested that ame-bantam-3p exerts an important function in larvaldevelopment. 20-Hydroxyecdysone (20E), a key steroid hormone, also regulates the expression of many miRNAs in insects. The transcriptional levels of miR-8-5p and miR-2a-3p in the planthopper *Nilaparvatalugens* [55]; miR-281 in *B. mori* [8]; and miR-8 and miR-14 in *Drosophila melanogaster* [56,57] were all reported to be inhibited in response to 20E treatment. Similarly, we found that the expression level of ame-bantam-3p was reduced in five-, six-, and seven-day-old larvae fed exogenous 20E. Meanwhile, we also measured the levels of 20E during larval–pupal developmental stages, and we found that the 20E titer was strongly and negatively correlated with ame-bantam-3p and *megf8* expression, remaining at a very low level in the larval stages and at a high level in the prepupal stage (Figure 7D). Using a 20E injection assay, we demonstrated that this hormone can repress the transcription of ame-bantam-3p and that of its target gene, *megf8*. Together, these results shed light on the molecular mechanism underlying the regulatory effect of 20E on ame-bantam-3p, as well as its pivotal role in insect development.

Some miRNAs are components of additional systemic signaling pathways in insects, including the 20E, insulin, stress, immune, and circadian pathways [41]. Bantam is involved in the regulation of insect metamorphosis by targeting *spook* and *spookier* in the ecdysteroid synthesis pathway [13,42]. Moreover, bantam promotes systemic growth by inhibiting the synthesis of the steroid hormone ecdysteroid [13]. In summary, we present evidence that both conserved and lineage-specific miRNAs participate in the regulation of metamorphosis in insects by directly controlling ecdysteroid biosynthesis. In our study, we found that the disruption of ecdysteroid biosynthesis through the use of ame-bantam-3p agomir led to a decrease in the expression levels of the *dib*, *phm*, *sad*, *nvd*, and *shd* genes, which encode enzymes required for ecdysteroid biosynthesis, whereas ame-bantam-3pantagomir injection increased the expression of these genes. Furthermore, 20E signaling was also reduced following ame-bantam-3pagomir treatment, as evidenced by the expression levels of genes coding for nuclear receptors (*ECRA*, *ECRB1*, and *USP*) and those of 20E primary response genes (i.e., the transcription factor-encoding genes *Br-C*, *E75*, and *E93*) (Figure 8). In addition, by employing RNA interference, we found that *megf8* is also involved in the regulation of ecdysteroid synthesis and 20E signaling pathways. These results showed that the expression of these 20E response genes is dependent on the effects of ame-bantam-3p on *megf8* expression. Meanwhile, ame-bantam-3p agomir treatment upregulated the expression of *megf8* and resulted in pupation failure and the promotion of cell proliferation and, consequently, body growth, which blocked larval metamorphosis. Bantam represses apoptosis directly by inducing the expression of *DIAP1* and indirectly by inducing the expression of *Yki*, which, in turn, suppresses the expression of the proapoptotic gene *hid* [16]. Additionally, we observed a greater number of proliferating fat body cells in ame-bantam-3p agomir-injected larvae than in larvae injected with NC agomir, as detected by EdU staining. The *dMegf8* gene is essential for *Drosophila* viability and the loss of its function results in lethality during larval stages, and the phenotype of *dMegf8* larval mutants is similar to that of some BMP signaling mutants, such as impaired formation of denticles and bristle [30]. In the present study, *megf8* knockout caused early pupation of larvae and significantly increased the expression levels of genes involved in ecdysteroid synthesis and 20E signaling-related genes in honeybee larvae (Figure 9). Accordingly, our data suggest that, as in *Drosophila*, ame-bantam-3p is involved in larval development by inhibiting 20E release. Ame-bantam-3p plays an important role in positively regulating the levels of the *megf8* gene, thus contributing to the homeostasis of 20E signaling pathway-associated genes during larval–pupal development in the honeybee. This study may provide novel insights into miRNA/gene interactions during metamorphosis in the honeybee.

## 4. Materials and Methods

### 4.1. In Vitrorearing of Honeybee Worker Larvae

One-day-old honeybee larvae were collected from colonies of the experimental apiary at Shandong Agricultural University (Tai’an, China). The queen bee was confined to a defined area on a comb containing empty cells for 12 h to control the age of the brood. Subsequently, the combs with newly laid eggs were moved to a separate place in the same colony using a queen excluder. After four days, the combs containing the synchronized two-day-old larvae (no more than 12 h old) were quickly transferred from the brood comb to 48-well culture plates and maintained in an incubator (temperature: 33 ± 1 °C; relative humidity: 55 ± 10%; in the dark). At the prepupal and pupal stages, the relative humidity was decreased to 80% and 70%, respectively. Daily diet volumes provided to the larvae and the approximate composition of the diet are detailed in Table 2.

For feeding, 20E (Sigma, St. Louis, MO, USA) was dissolved in 100% ethanol (50 mg/mL) to prevent crystallization of the steroid and then diluted in Ringer solution containing 130 mM NaCl, 6 mM KCl, 4 mM MgCl_2_, 5 mM CaCl_2_, 160 mM sucrose, 25 mM glucose, and 10 mM HEPES (pH 6.7, 500 mOsmol) to give a final concentration of 10 mg/mL 20E. For the treatment group, larvae were fed a diet containing 20E (0.01 mg/mL). Control larvae were fed the normal diet with the corresponding doses of ethanol diluted in Ringer solution as the solvent control.

### 4.2. MicroRNA Sequencing

For small RNA sequencing, samples from five-day-old larvae, six-day-old (last-instar) larvae, seven-day-old larvae (prepupae), eight-day-old larvae (prepupae), and 10-day-old larvae (white-eye pupae, Pw) from different treatments were collected and, respectively, pooled. The samples were collected at least three times during different rearing cycles at each developmental stage and from each treatment group, thus serving as biological replicates. The mixed samples were immediately frozen in liquid nitrogen and stored at −80 °C for RNA extraction and small RNA library construction. Total RNA was extracted from each sample using TRIzol reagent (TaKaRa, Dalian, China) according to the manufacturer’s instruction. RNA concentrations were measured using a Qubit RNA Assay Kit and a Qubit 2.0 Fluorometer (Life Technologies, Santa Clara, CA, USA). RNA integrity was assessed using an Agilent 2100 Bioanalyzer (Agilent, Santa Clara, CA, USA). Total RNA was stored at −80 °C for subsequent sequencing. RNA libraries were constructed and generated using a NEBNext Ultra II small RNA Sample Library Prep Kit for Illumina (NEB, San Diego, CA, USA) following the manufacturer’s recommendations, then they were sequenced using the Illumina HiSeq.2500 Platform. Sequencing was undertaken by Biomarker Technologies Company (Beijing, China). Clean data (clean reads) were obtained by removing reads containing adapter, reads containing ploy-N, and low-quality reads from raw data. The clean reads were, respectively, mapped to the SILVA, GtRNAdb, Rfam, and Repbase databases. Sequence alignment was undertaken using the Bowtie tool [58] to filter out ribosomal RNAs (rRNAs), transfer RNAs (tRNAs), small nuclear RNAs (snRNAs), small nucleolar RNAs (snoRNAs), other ncRNAs, and repeats. The remaining reads were used to detect novel miRNAs predicted via a comparison with the honeybee genome (*Apis_mellifera*. Amel_4.5_ncbi) and known miRNAs from miRBase (v22) [59]. Randfold was used for the prediction of novel miRNA secondary structure. Differential expression analysis between two conditions/groups was performed using the “DESeq2” R package (1.10.1) [44], which provides statistical routines for determining differential expression in digital miRNA expression data using a model based on the negative binomial distribution. The resulting *p*-values were adjusted using the Benjamini and Hochberg approach for controlling the false discovery rate. MiRNAs with |log2(FC)| ≥ 0.58 and a *p*-value ≤ 0.05 were assigned as differentially expressed.

### 4.3. MiRNA Target Gene Prediction and Functional Analysis

Based on the sequences of the miRNAs (known and novel), miRanda [45] and targetScan [46] were used to identify the targets of miRNAs and run by default parameters. The final targets were intersected with the gene sets identified by two softwares. The sequences of the predicted target genes were compared with those of the non-redundant (NR) [60], Swiss-Prot [60], Gene Ontology (GO) [61], Clusters of Orthologous Groups of Proteins (COG) [62], Kyoto Encyclopedia of Genes and Genomes (KEGG) [63], Eukaryotic Orthologous Groups (KOG) [64], and Pfam [65] databases using *BLASTX* searches with an E-value cutoff of <1.00 × 10^−5^ to obtain annotation information. 

### 4.4. Quantitative PCR for miRNA and mRNA

Total RNA was extracted using TRIzol reagent (TaKaRa) following the manufacturer’s instructions. First-strand cDNA for mRNA and miRNA was synthesized using, respectively, *Evo M-MLV* Premix and a miRNA 1st strand cDNA Synthesis Kit (Accurate Biotechnology Co., Ltd, ChangSha, China) according to the manufacturer’s protocols. All the primers for RT-qPCR were designed and synthesized by Sangon Biotechnological Co. (Shanghai, China). RT-qPCR was conducted on a 7500 Real-time PCR System (ABI, USA) using *TransStart®* Top Green qPCR SuperMix (TransGen Biotech, Beijing, China) according to the manufacturer’s instructions. The sequences of all the primers used for PCR are listed in Table 3. Each treatment had at least three biological repeats with three technical repeats. *β-actin* (GenBank accession no. XM_017065464) and *U6* were used as the reference genes for mRNA and miRNA, respectively. Relative gene expression levels were calculated using the 2^−ΔΔCt^ method [66].

### 4.5. Cell Transfection and Dual-Luciferase Reporter Assay

The pGL3-CMV-LUC vector (Genomeditech, Shanghai, China) was used as a firefly luciferase reporter vector and the fragment tested covering the binding site of the miRNA to the target gene of roughly 400 bp were cloned downstream of the firefly luciferase gene. HEK-293 cells were seeded into 24-well plates with 500 µL of DMEM (Gibco, Carlsbad, CA, USA). At approximately 70% confluency, the cells were transfected as follows. First, 100 ng of plasmid DNA and 30 nM ame-bantam-3p mimics (5′-TGAGATCATTGTGAAAGCTGATT-3′) (GeneBiogist, Shanghai, China) were dissolved in 50 µL of Opti-MEM (Invitrogen, Carlsbad, CA, USA), 3 µL of Lipofectamine (Invitrogen) was dissolved in 50 µL of Opti-MEM, and the two solutions were mixed together and incubated at room temperature for 30 min. At 48 h post-transfection, cell lysates were prepared and assayed for firefly luciferase activity using a Dual-Luciferase Reporter Assay Kit (Promega, Madison, WI, USA) according to the manufacturer’s protocol. Luciferase activity was assessed using an Infinite M1000 microplate reader (Tecan, Switzerland). Each experiment was performed in triplicate. The mean of the relative firefly luciferase/*Renilla* luciferase expression ratio of the control was set to 1. The data were analyzed with two-tailed *t*-tests.

### 4.6. Injection of Ame-Bantam-3p Agomir and Antagomir

Ame-bantam-3p and negative control (NC) agomir and antagomir were synthesized by Sangon Biotechnological Co. After dilution to 1 nmol/μL, 3 μL of miRNA agomir and antagomir or negative control (NC) agomir and antagomir were injected into the hemolymph of six-day-old larvae, respectively, using a microsyringe. A total of 24 larvae were injected with each miRNA agonist and antagomir. After 24 h, some of the larvae (four biological replicates) were dissected and stored at −80 °C for further analysis, and the remaining larvae were used for phenotypic analysis and determination of survival.

### 4.7. 20Etiter Determination

To estimate 20E titers, 100 μL of hemolymph was collected from no less than 10 agomir-injected or antagomir-injected larvae 24 and 36 h post-injection. A 20E enzyme-linked immunosorbent assay (ELISA) kit (Meimian Industrial Co., Ltd., Jiangsu, China) was used to determine the 20E titer following the manufacturer’s instructions. The experiment was performed with three biological replicates.

### 4.8. 5-Ethynyl-2′-Deoxyuridine (EdU) Incorporation Assay

Cell proliferation assays were undertaken using the BeyoClick EdU Cell Proliferation Kit with Alexa Fluor 488 (Beyotime Biotechnology, ShangHai, China) according to the manufacturer’s instructions. The solution-injected larvae were injected with 10 ngof EdU and the larval fat bodies were dissected out 12 h after injection. The samples were subsequently fixed in 4% paraformaldehyde for 12 h and were transversely sliced into 7-μm-thick sections on a freezing microtome. After washing three times in PBS, the sections were permeabilized with 0.5% Triton X-100 at room temperature for 15 min. The fixative was then removed, and the sections were washed first with PBS containing 3% BSA and then three times with PBS. Subsequently, the sections were incubated in Click Additive Solution in the dark and stained with DAPI. Fluorescence images of EdU incorporation were then obtained under Leica TCS SPE confocal microscope (Carl Zeiss, Oberkochen, Germany).

### 4.9. Terminal Deoxynucleotidyl Transferase-Mediated dUTP-Biotin Nick End Labeling (TUNEL) Assay

The fat bodies of injected worker bee larvae were harvested, fixed in 4% paraformaldehyde for 12 h, and then transversely sliced into 7-μm-thick sections on a freezing microtome. The One Step TUNEL Apoptosis Assay Kit from Beyotime was used for the detection of programmed cell death (PCD) in the tissues based on the manufacturer’s instructions. The sections were subsequently incubated with 0.5% Triton X-100 on a shaker at room temperature for 15 min, washed three times in PBST, incubated with 50 μL of TUNEL inspection fluid for 1 h at 37 °C, and counterstained with DAPI at room temperature for 10 min. After washing three times with PBST, the stained sections were observed and imaged using Leica TCS SPE confocal microscope (Carl Zeiss, Oberkochen, Germany).

### 4.10. Statistical Analysis

All data were expressed as means ± standard error of the mean (SEM) derived from at least three biological repeats. Student’s *t*-tests (two groups) or one-way ANOVA followed by Dunnett’s post hoc test (three or more groups) was employed for comparisons using GraphPad Prism 8.0 (GraphPad Software, San Diego, CA, USA). Differences were considered significant at *p* < 0.05.

## 5. Conclusions

In conclusion, high ame-bantam-3p activity in young larvae was found to promote the expression of its target gene *megf8*, thus contributing to the maintenance of low 20E titers and the promotion of systemic growth. In contrast, reduced ame-bantam-3p activity contributed to the generation of the 20E peak, the cessation of growth, and entry into metamorphosis (Figure 10).

## Figures and Tables

**Figure 1 ijms-24-05726-f001:**
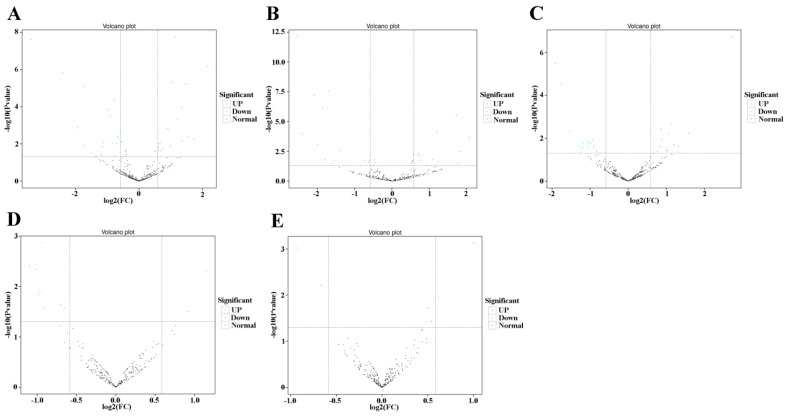
Volcano plot of DE miRNAs in (**A**) R5d vs. E5d; (**B**) R6d vs. E6d; (**C**) R7d vs. E7d; (**D**) R8d vs. E8d; and (**E**) R-Pw vs. E-Pw.

**Figure 2 ijms-24-05726-f002:**
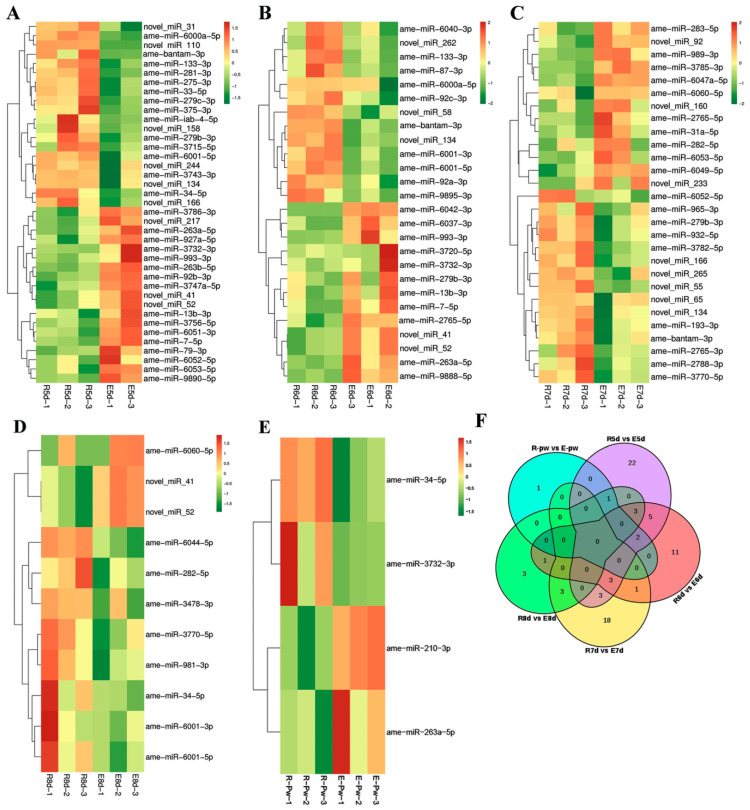
20-Hydroxyecdysone (20E)-responsive miRNAs identified by small RNA sequencing. Hierarchical cluster analysis in (**A**) R5d vs. E5d; (**B**) R6d vs. E6d; (**C**) R7d vs. E7d; (**D**) R8d vs. E8d; and (**E**) R-Pw vs. E-Pw. The green and red rectangles indicate downregulated and upregulated miRNAs, respectively. Venn plot (**F**) of differentially expressed (DE) miRNAs among all examined samples. Numbers represent the identified DE miRNAs in pairwise comparisons.

**Figure 3 ijms-24-05726-f003:**
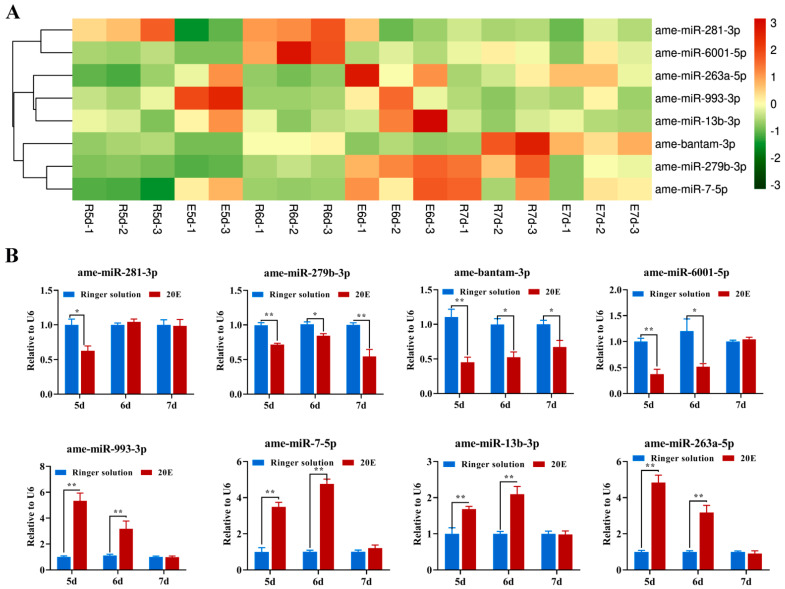
Validation of eight selected differentially expressed (DE) miRNAs by RT-qPCR. (**A**) Hierarchical cluster analysis of eight selected DE miRNAs at different developmental stages in honeybees. (**B**) Determination of eight selected DE miRNAs by RT-qPCR. Three biological replicates and three technical replicates were performed. * *p* < 0.05, ** *p* < 0.01 (Student’s *t*-test). The values are means ± SEM.

**Figure 4 ijms-24-05726-f004:**
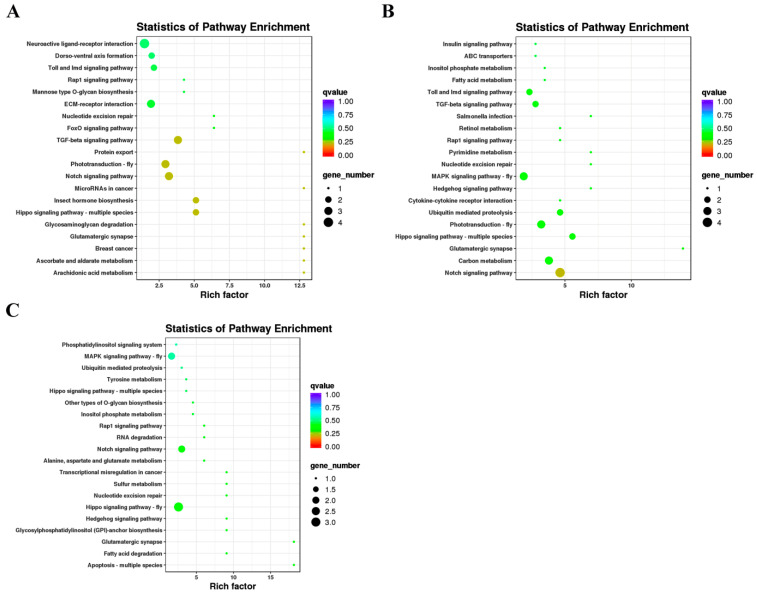
KEGG pathways enrichment analysis of target genes in (**A**) R5d vs. E5d; (**B**) R6d vs. E6d; and (**C**) R7d vs. E7d.

**Figure 5 ijms-24-05726-f005:**
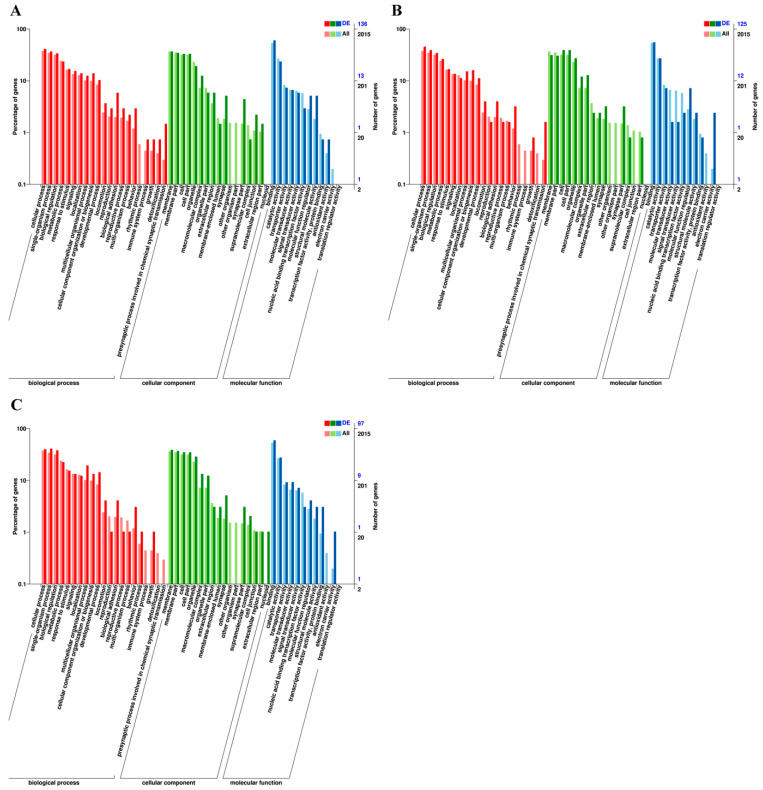
Gene ontology (GO) term enrichment analysis of target genes in (**A**) R5d vs. E5d; (**B**) R6d vs. E6d; and (**C**) R7d vs. E7d.

**Figure 6 ijms-24-05726-f006:**
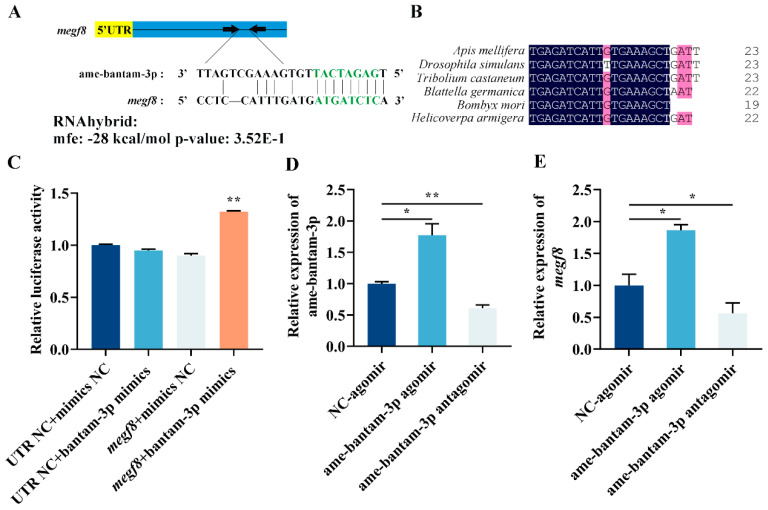
In vitro dual-luciferase reporter assays of miRNA–mRNA interactions in worker bee larvae. (**A**) Ame-bantam-3pbinding sites in the megf8 gene as predicted by RNAhybrid. The blue area is the coding sequence (CDS) and the yellow area is the untranslated region (UTR). The seed sequence and complementary bases are highlighted in green. (**B**) Multiple alignment of ame-bantam-3p in insects and *Apis mellifera*. (**C**) Relative luciferase activity in the dual-luciferase assay. UTR NC represents a reporter without the target sequence and mimics NC represents mimic controls. The mean ± SEM of the relative luciferase expression ratio (firefly luciferase/Renilla luciferase, Luc/R-luc) was calculated from three biological replicates (Student’s *t*-test, n = 3) and compared with the negative control (NC). The expression level of ame-bantam-3p (**D**) and *megf8* (**E**) in larvae injected with ame-bantam-3p agomir, ame-bantam-3p antagomir or NC. Reverse transcription quantitative polymerase chain reaction (RT-qPCR) data are presented as means ±SEM of triplicate samples. * *p* < 0.05, ** *p* < 0.01 (Student’s *t*-test).

**Figure 7 ijms-24-05726-f007:**
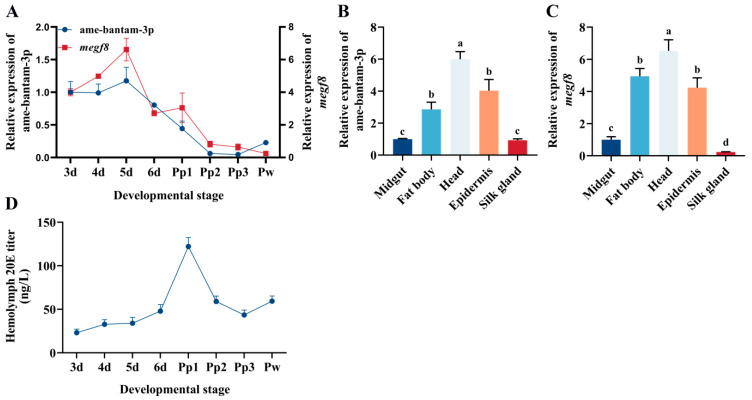
(**A**) The expression levels of ame-bantam-3p and *megf8* in the larval–pupal developmental stages. The expression levels of ame-bantam-3p (**B**) and *megf8* (**C**) in isolated tissues. Vertical bars represent the mean ± SEM (n = 5) and letters represent significant differences (*p* < 0.05). Relative expression levels were denoted as fold-change over that of the internal controls *U6* and *β-actin*, respectively. Each point and histogram bar represents the relative expression, the error bars indicate SEM. (**D**) Hemolymph 20-hydroxyecdysone (20E) titers in larval–pupal developmental stages of the honeybee. Each point indicates the mean ecdysone concentration ± SEM.

**Figure 8 ijms-24-05726-f008:**
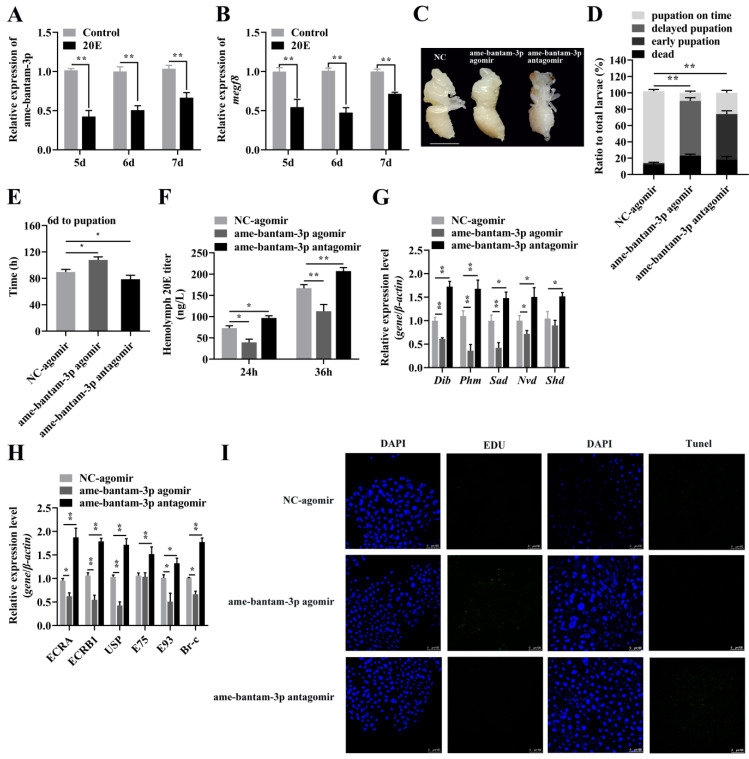
The effects of exogenous 20-hydroxyecdysone (20E) application on the expression profiles of ame-bantam-3p (**A**) and *megf8* (**B**) five, six, and seven days after feeding. (**C**) Phenotypeassessment after ame-bantam-3p agomir or ame-bantam-3p antagomir injection. The ruler represents 0.3 cm. (**D**) Percentage distribution of different phenotypes. (**E**) Statistical analysis of the time of development of six-day old larvae to pupae. (**F**) The determination of the 20E titer 24 and 36 h after miRNA agomir or antagomir injection. The expression levels of genes essential for ecdysteroid synthesis (**G**) and 20E signaling pathway genes (**H**) 24 h after ame-bantam-3p agomir or ame-bantam-3p antagomir injection. (**I**) Representative image of fat body cell proliferation and apoptosis. The EdU assay was performed 24 h after agomir or antagomir injection into the hemocoel. TUNEL staining showing the morphology of the fat body. Bar indicates 100 μm. Error bars indicate the SEM.* *p* < 0.05, ** *p* < 0.01 compared with the relevant control (two-tailed *t*-test).

**Figure 9 ijms-24-05726-f009:**
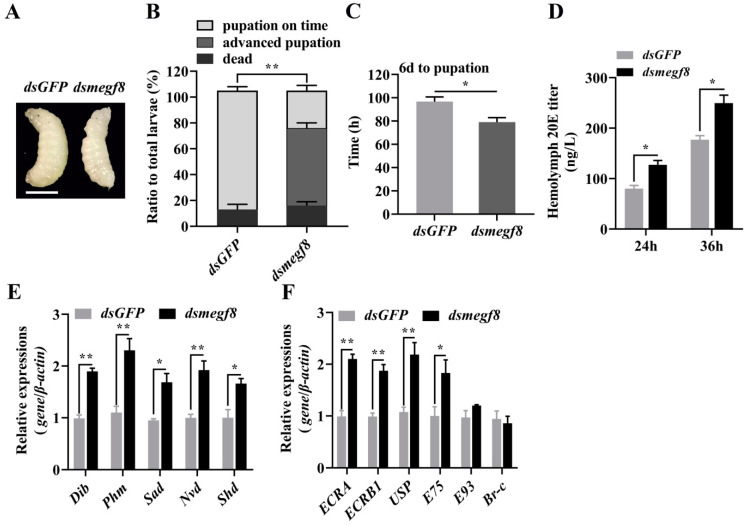
Injection of dsRNA from *megf8* by the six-day old larvae inhibits ecdysteroid synthesis and represses 20E signal in *A*. *mellifera*. (**A**) Phenotypes after *dsmegf8* injection; the ruler represents 0.3 cm. (**B**) Percentage distribution of different phenotypes. (**C**) Statistical analysis of the time of development of six-day old larvae to pupae. (**D**) The determination of the 20E titer 24 and 36 h after *dsmegf8* injection. The expression levels of genes essential for ecdysteroid synthesis (**E**) and 20E signaling pathway genes (**F**) 24 h after *dsmegf8* injection. * *p* < 0.05, ** *p* < 0.01 (Student’s *t*-test). Error bars indicate the mean ± SEM (n = 5).

**Figure 10 ijms-24-05726-f010:**
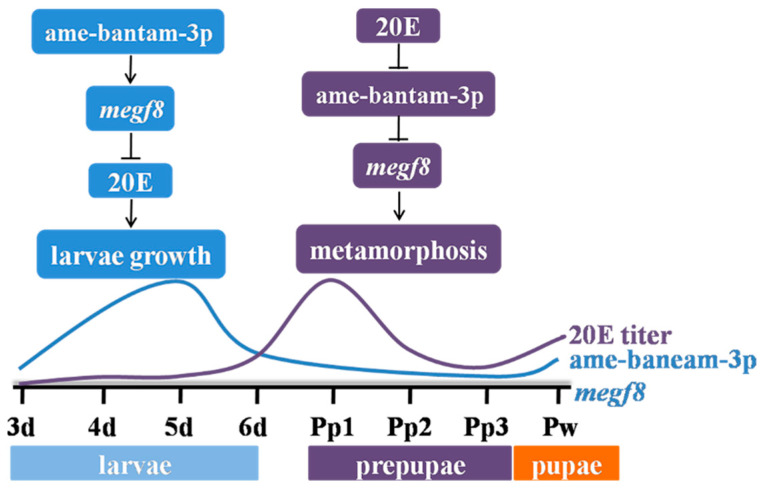
Mechanism explaining the functions of ame-bantam-3p and *megf8* during larval–pupal development in the honeybee.

**Table 1 ijms-24-05726-t001:** miRNA count among different tissues.

DEG Set	DEG Number	Up-Regulated	Down-Regulated
R5d-1_R5d-2_R5d-3 vs. E5d-1_E5d-2_E5d-3	39	19	20
R6d-1_R6d-2_R6d-3 vs. E6d-1_E6d-2_E6d-3	26	13	13
R7d-1_R7d-2_R7d-3 vs. E7d-1_E7d-2_E7d-3	28	13	15
R8d-1_R8d-2_R8d-3 vs. R8d-1_R8d-2_R8d-3	11	3	8
R-Pw-1_ R-Pw-2_ R-Pw-3 vs. E-Pw-1_ E-Pw-2_ E-Pw-3	4	2	2

Pw indicate the white eye pupa.

**Table 2 ijms-24-05726-t002:** Composition of larvae diets.

Ingredients	Content (%)
Royal jelly	50.00
Glucose	6.00
Fructose	6.00
Yeast extract	1.00
Sterile water	37.00
Total	100.00

**Table 3 ijms-24-05726-t003:** The primer sequences used for qPCR and construction of luciferase reporter vector.

Gene	Application	Sequence (5′-3′)
U6-F	qRT-PCR	ACACTCCAGCTGGGTCAAAATCGTCAAGCG
U6-R ame-bantan-3p		CTCAACTGGTGTCGTGGAGTCGGCAA TGAGATCATTGTGAAAGCTGATT
Ame-miR-281-3p		TGTCATGGAGTTGCTCTCTTTGT
Ame-miR-279b-3p		TGACTAGATCGAAATACTCGTCCC
Ame-miR-6001-5p		GTAGGTAACGACTGATGGGAAC
Ame-miR-993-3p		GAAGCTCGTCTCTACAGGTATCT
Ame-miR-7-5p		TGGAAGACTAGTGATTTTGTTGT
Ame-miR-13b-3p		TATCACAGCCATTTTTGACGATT
Ame-miR-263-5p		GTAAATGGCACTGGAAGAATTCAC
*Megf8*-F		GGCTATTACGGCGATCCAAGAGATG
*Megf8*-R		CTGCCAAGACCTTGTTTGCCATTTC
*ECRA*-F		GAAGTGTTTGACGGTCGGGATGAG
*ECRA*-R		GCCTTCTTCTCCTTCCGCTTCAC
*ECRB1*-F		CCTGGCTCTTTGAACGGGTATGG
*ECRB1*-R		CCTCCTCCTCCTCCTCCTCCTC
*USP*-F		CACTGGACATGAAGCCCGACAC
*USP*-R		AGGGTGCGACTGCTTTGTTCTG
*E93*-F		ACGACGACGACGACGACTACG
*E93*-R		GCGGTGGATTGGAGATGGTGATG
*E75*-F		ACGAGTAGTACGAGCACGAGTCC
*E75*-R		ACTGTCCGCTATGTCCACCTGTAG
*Br-c*-F		GGTCAGCCATAAGAGCCAGTATCAC
*Br-c*-R		CCATCGGTGTCTGCCAATACTTCG
*Dib*-F		GGACATTGTGGGCATGGCTT
*Dib*-R		TGTGTGGCTTCGATCCTCAATT
*Phm*-F		ATCCGCAACTGATACGCCAATCG
*Phm*-R		TCCTCTGATCCTTCCACTGTTCCC
*Sad*-F		TCATCCGCCGTAACTCCAGGTATC
*Sad*-R		GCTACCCGAAGATCGTGCCATG
*Nvd*-F		CGTTCCTCATGCCTCTTCACCAC
*Nvd*-R		ACCACGTCGTTCCAACTGTTCAC
*Shd*-F *Shd*-R		CGCATCGCAGACATCAAGAATCG ACTTTCGCTAGTTGCTACCGCTAC
*β-actin*-F *β-actin*-R		TTATATGCCAACACTGTCCTTT AGAATTGATCCACCAATCCA
*Megf8*-ame-bantan-3p-F *Megf8*-ame-bantan-3p-R	Luciferase reporter assay	AAGATCGCCGTGTGACTCGAGCA CGCCCCGACTCTAGCACGC
*dsMegf8*-F *dsMegf8*-R	dsRNA	TAATACGACTCACTATAGGGCGAGGGAACAGAATGTAGCTATG TAATACGACTCACTATAGGGCGATGTGCTGCCCGTGGCCTTAA

Underlined sequences indicate the T7 adaptor; F, forward primer; R, reverse primer.

## Data Availability

Data is contained within the article or Appendix A.

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
