# Peer review of "The MicroRNA Ame-Bantam-3p Controls Larval Pupal Development by Targeting the Multiple Epidermal Growth Factor-like Domains 8 Gene (megf8) in the Honeybee, Apis mellifera"

_ijms, 2023, doi:10.3390/ijms24065726_

Round 1

Reviewer 1 Report

The miRNA work is very comprehensive and combines multiple disciplines from miRNA sequencing to bioinformatics analysis and is validated through functional assays. 

My main concern is the log2(FC) value you chose as the cutoff. Why is the chosen value very low? Can you explain and justify the used cutoff?

Other than that, please proofread the text again, there are many typing errors throughout the manuscript. I have marked them red in the pdf attached. 

The supplementary figures are also of low quality and hard to read. Please improve on that. The captions are also not sufficient in detail. Please improve the caption as well. 

Author Response

Dear reviewer:

Thank you for your decision and constructive comments on my manuscript. We have carefuly considered the suggestion of Reviewer and make some changes according to your comments. Revision notes, point-to-point, are given as follows:

The miRNA work is very comprehensive and combines multiple disciplines from miRNA sequencing to bioinformatics analysis and is validated through functional assays.

My main concern is the log2(FC) value you chose as the cutoff. Why is the chosen value very low? Can you explain and justify the used cutoff?

|log2(FC)|≥0.58 , the FC value is 1.5, and the general FC value for mRNA is 2, because there are not as many predicted small RNAs as mRNAs, and the conditions for differential screening will be a little more lenient. MiRNAs with |log2(FC)|≥0.58 and a P-value≤0.05 were assigned as differentially expressed. Some small RNAs are supported by the literature, e.g. Su et al., 2018, Xu et al., 2020.

Other than that, please proofread the text again, there are many typing errors throughout the manuscript. I have marked them red in the pdf attached.

We have proofread the text and carefully corrected any errors in the text.

The supplementary figures are also of low quality and hard to read. Please improve on that. The captions are also not sufficient in detail. Please improve the caption as well.

We have revised the supplementary figures. 

Author Response

Dear reviewer:

Thank you for your decision and constructive comments on my manuscript. We have carefuly considered the suggestion of Reviewer and make some changes according to your comments. Revision notes, point-to-point, are given as follows:

Title: The current title requires a lot of specialised knowledge to understand it and therefore does not attract the reader. I would suggest ‘The microRNA ame-bantam-3p controls larvalpupal development by targeting the Multiple Epidermal Growth Factor-like Domains 8 gene (megf8) in the honey bee, Apis mellifera’

We had revised the title according to the reviewers' suggestions.

Ecdysone: ‘ecdysone’ is misused throughout this manuscript. The major hormonal steroid in insects is 20-hydroxyecdysone, which is a member of a class of steroids correctly referred to as ‘ecdysteroids’ and which interacts with the ‘ecdysteroid receptor’ complex. Ecdysone is a specific compound which has rather low affinity for the ecdysteroid receptor. The authors need to examine each case where they have used ‘ecdysone’ in the text and decide if they really meant ‘ecdysteroid’, ‘20-hydroxyecdysone (20E)’ or ‘ecdysone’.

We had revised the ‘ecdysone’ in the text according to the reviewers' suggestions.

P2, line 49: Blatella germanica is the German cockroach

We had revised the German flatworm for German cockroach. (line 53)
P2 line 64: shouldn’t it be ‘dmMegf8’ (dm for Drosophila melanogaster)

The reference we cite was described as dMegf8
P3 line 116: MgCl2 and CaCl2

We had revised MgCl and CaCl for MgCl2 and CaCl2. (line 121)
P3 line 122: which instar are 5d-larvae in? What age in days are white-eyed pupae?

The 10-day-old was white-eyed pupae. (line 128)
P5 line 198: 100μL haemolymph – from how many larvae?

Hemolymph was collected from no less than 10 larvae. (line 207)
P5 line 237: the font is not uniform

We had revised the font. (line 247)
P5 line 243: are these percentages really accurate to 2dp?

These percentages were really accurate to 2dp.
P6 line 246: 18 and 30 correspond to the limits the authors set!

The raw sequences obtained from sequencing contain adapter sequences or low quality sequences, in order to ensure the accuracy of information analysis, it is necessary to quality control the raw data to obtain high quality sequences (clean reads):

(1) Removal of adapter.

(2) Removal of sequences shorter than 18 nt or longer than 30 nt.

(3) For each sample, sequences with low quality values are removed.

(4) Removal of Reads with ploy-N (N is an unidentifiable base) content greater than or equal to 10%.
P7 line 304: explain the abbreviation DE

DE indicated differential expression.
P7 line 336: In addition,

We had revised In additional for In addition. (line 348)
P9 line 420: define PCD

PCD was defined as Programmed cell death. (line 435)
P11 line 520: define BMP

BMP was defined as Bone morphogenetic protein. (line 68)
References: species names should be italicized

We had revised species names in references.
Table 1: Composition of larval diet; Total

We had revised for Toal for Total. (line 709)
Figure 2: ‘triangles’ should be ‘rectangles’

We had revised for triangles for rectangles. (line 721)
Figure 3: the legend refers to Panels A and B, but only ‘A’ appears on the figure, and the graphs seem to correspond to ‘B’

We had revised the figure. (line 724)
Figure 4: far too small! The panels are illegible

We had revised the figure.
Figure 6D: the units for 20E titre are given in ‘μL/mL’ which cannot be correct

The unit for 20E titer was ng/L. (line 748)
Figure 8D: the units for 20E haemolymph titre are given in ug/mg. The highest value is ca.250 μg/mg, which corresponds to ca. 0.5M; this cannot possibly be correct!

The unit for 20E titer was ng/L. (line 768)
Figure S3: more information is required in the legend to explain this figure

We had added some information in the legend to explain this figure.
Figure S4: more explanation in the legend; honeybee larval samples; ‘triangles’ should be ‘rectangles’

We had revised the triangles for rectangles.

Round 2

Reviewer 2 Report

Revised Version

The authors have dealt, at least in part, with most of the comments and modified the manuscript accordingly. However, it is a shame that they did not properly proof-read the modified manuscript after making the changes and corrections as they would have realised that, having changed the figure numbering,  the text no longer reflects this. Also, there are still many places where spaces have not been left between words.

The misuse of ‘ecdysone’ has been corrected in certain instances but not in others. Taking the Introduction as an example, the following are still wrong:

Line 55: 20E (I know that the authors of ref [13] used ‘ecdysone’, but they actually measured 20-hydroxyecdysone)

Line 72: 20E

Line 74: 20E, the active ecdysteroid,

Line 75: ecdysteroid receptor

Line 81: the regulation of ecdysteroid and JH levels

Line 87: ecdysteroid

Line 89: 20-hydroxyecdysone

The authors need to make similar changes throughout the Discussion.

Corrections to Abstract (as this is the Section which will be most widely read)

Line 14: in different (if one is ‘indifferent’ to something, one does not care about it)

Line 18: -3p is

Line 25: change ‘significant’ to ‘significantly’

Line 30: change ‘indicated’ to ‘indicate’

Author Response

The misuse of ‘ecdysone’ has been corrected in certain instances but not in others. Taking the Introduction as an example, the following are still wrong:

Line 55: 20E (I know that the authors of ref [13] used ‘ecdysone’, but they actually measured 20-hydroxyecdysone)

We had revised ecdysone for 20E. (Line 55)

Line 72: 20E

We had revised ecdysone for 20E. (Line 72)

Line 74: 20E, the active ecdysteroid,

We had revised ‘20E, the active form of ecdysone’ for ‘20E, the active ecdysteroid’. (Line 74)

Line 75: ecdysteroid receptor

We had revised ‘ecdysone receptor’ for ‘ecdysteroid receptor’. (Line 75)

Line 81: the regulation of ecdysteroid and JH levels

We had revised ‘ecdysone’ for ‘ecdysteroid’. (Line 81)

Line 87: ecdysteroid

We had revised ‘ecdysone’ for ‘ecdysteroid’. (Line 87)

Line 89: 20-hydroxyecdysone

We had revised ‘ecdysone’ for ‘20-hydroxyecdysone’. (Line 89)

The authors need to make similar changes throughout the Discussion.

 We had revised the Discussion.

Corrections to Abstract (as this is the Section which will be most widely read)

Line 14: in different (if one is ‘indifferent’ to something, one does not care about it)

We had revised ‘indifferent’ for ‘in different’. (Line 14)

Line 18: -3p is

We had revised ‘-3pis’ for ‘-3p is’. (Line 18)

Line 25: change ‘significant’ to ‘significantly’

We had revised ‘significant’ for ‘significantly’. (Line 25)

Line 30: change ‘indicated’ to ‘indicate’

We had revised ‘indicated’ for ‘indicate’. (Line 30)
